# A Comprehensive Review of Essential Oil–Nanotechnology Synergy for Advanced Dermocosmetic Delivery

Redouane Achagar [1], Zouhair Ait-Touchente [2,*], Rafika El Ati [3], Khalid Boujdi [4], Abderrahmane Thoume [1], Achraf Abdou [1] and Rachid Touzani [3]

1 Laboratory of Organic Synthesis, Extraction and Valorization, FSAC, Hassan II University of Casablanca, Maarif, B.P. 2693, Casablanca 20000, Morocco; redouane.achagar-etu@etu.univh2c.ma (R.A.); thoumecv@gmail.com (A.T.); achraf.abdou-etu@etu.univh2c.ma or abdou.achraf01@gmail.com (A.A.)
2 Universite Claude Bernard Lyon-1, Centre National de la Recherche Scientifique (CNRS), ISA-UMR 5280, 69100 Villeurbanne, France
3 Department of Chemistry, Laboratory of Applied Chemistry and Environment (LCAE), Faculty of Sciences, University Mohammed Premier, Oujda 60000, Morocco; rafika.elati@gmail.com (R.E.A.); r.touzani@ump.ac.ma (R.T.)
4 Faculty of Sciences and Technologies Mohammedia, University Hassan II, B.P. 146, Mohammedia 28800, Morocco; kh.boujdi@gmail.com
* Correspondence: zouhair.ait-touchente@univ-lyon1.fr

**Abstract:** This review investigates the convergence of nanotechnology and essential oils in advanced dermocosmetic delivery. It outlines the pivotal role of inorganic and polymeric nanoparticles, such as titanium dioxide, zinc oxide, and gold nanocarriers, in cosmeceutical applications, facilitating slow release, deeper skin penetration, and increased retention of active compounds. Essential oils, renowned for therapeutic benefits, face translation challenges due to volatility and low water solubility. This review explores the potential use of plant nanovesicles as carriers, emphasizing safety, stability, and scalability, offering a sustainable and cost-effective industrial application. Nanomaterial integration in consumer products, particularly cosmetics, is prevalent, with nanocarriers enhancing the permeation of bioactive compounds into deeper skin layers. The review emphasizes recent nanotechnological advancements, covering nanoparticle penetration, experimental models, and therapeutic applications in dermatology, ranging from non-invasive vaccination to transdermal drug delivery. Additionally, the review delves into nanomaterials' role in addressing skin aging, focusing on tissue regeneration. Nanomaterials loaded with cosmeceuticals, such as phytochemicals and vitamins, are explored as promising solutions to mitigate signs of aging, including wrinkles and dry skin, providing innovative approaches to skin rejuvenation. Overall, the review offers a comprehensive synthesis of essential oil–nanoparticle synergy, shedding light on the current landscape and future potential of advanced dermocosmetic delivery systems.

**Keywords:** dermocosmetic delivery; nanotechnology; essential oils; nanoparticles; cosmeceutical applications

## 1. Introduction

In recent years, dermocosmetic research has experienced a noteworthy convergence of nanotechnology and essential oils, marking a transformative era in advanced delivery systems [1–5]. This intersection offers a promising solution to enduring challenges faced by essential oils in cosmetics, such as volatility and low water solubility [6–8]. Cutting-edge advancements in cosmeceuticals heavily depend on polymeric nanoparticles and inorganic substances such as titanium dioxide, zinc oxide, and gold nanocarriers. These particles enable gradual release, enhanced skin penetration, and prolonged retention of active compounds [7,9–13].

Essential oils, celebrated for therapeutic virtues, encounter challenges when incorporated into cosmetics due to volatility and limited water solubility [14]. Despite benefits

like antioxidant and anti-inflammatory properties, translating essential oils to cosmeceutical prominence requires strategic solutions. Nanotechnology provides a transformative platform, utilizing nanoparticles to augment the effectiveness of essential oils in dermatological applications [15–18].

Titanium dioxide, known for broad-spectrum UV protection; zinc oxide with multifunctional properties; and pliable gold nanocarriers represent pioneers in cosmeceutical frontiers [19–22]. These nanoparticles bring functionalities like slow and sustained release of active compounds, facilitated skin penetration, and prolonged retention within the skin matrix, ensuring a targeted and enduring impact of bioactive components [23–25].

Essential oils play a pivotal role in the formulation of cosmetic products due to their multifaceted benefits and complex composition of active compounds [14,17]. Derived from various plant sources through methods such as steam distillation, expression, and solvent extraction, these oils harbor a diverse array of chemical constituents that contribute to their distinct aromatic properties and therapeutic potential [26,27]. In cosmetics, essential oils serve as natural preservative agents, offering antimicrobial properties that safeguard against bacterial and fungal contamination, thereby enhancing the shelf life and stability of cosmetic formulations [28,29]. Additionally, their incorporation into skincare products brings about a spectrum of dermatocosmetic benefits, including anti-acne, anti-aging, skin lightening, and sun protection effects [30]. Furthermore, essential oils contribute to the olfactory experience of cosmetic products, imparting pleasing fragrances that appeal to consumers while also offering potential aromatherapeutic effects [31,32]. Despite the widespread use of synthetic fragrances in the industry, the rising demand for natural alternatives underscores the preference for essential oils due to their perceived safety and numerous health benefits [14]. However, it is essential to acknowledge the potential contraindications and allergic effects associated with their use, highlighting the importance of cautious formulation and consumer education [33]. Moreover, sustainable sourcing and cultivation practices are imperative to mitigate the environmental impact of large-scale harvesting, ensuring the conservation of biodiversity and protection of endangered plant species while meeting the growing demand for these valuable botanical ingredients in the cosmetic industry [14].

In response to challenges faced by essential oils, a green revolution is unfolding through the integration of plant nanovesicles into nanotechnological frameworks. Plant nanovesicles, characterized by lipid bilayer structures, emerge as promising carriers for essential oils, enhancing stability, safety, and efficacy. Derived from plant sources, these nanovesicles align with the growing trend towards sustainable and eco-friendly cosmetic formulations. Their ability to encapsulate and deliver essential oils in a controlled manner addresses volatility and solubility issues, providing a foundation for sophisticated dermocosmetic delivery systems [34–36].

The increasing integration of nanotechnology into consumer products, particularly cosmetics, is shaping skincare innovations significantly [7,37]. Nanocarriers, capable of penetrating deeper skin layers, enhance the bioavailability and efficacy of active compounds. This integration has led to substantial advancements, ranging from understanding nanoparticle penetration mechanisms to exploring experimental models and therapeutic applications in dermatology [38–40]. Nanotechnological strides are evident in non-invasive vaccination strategies and the facilitation of transdermal drug delivery, which are reshaping contemporary skincare practices [41,42]. A critical aspect of the nanomaterial revolution in dermocosmetics lies in its impact on addressing skin aging and promoting tissue regeneration [43–45]. Nanomaterials, enriched with cosmeceuticals such as phytochemicals and vitamins, offer innovative solutions to combat signs of aging, targeting concerns like wrinkles and dry skin. Through the utilization of nanotechnology, these dermocosmetic formulations present a multifaceted approach to skin rejuvenation, providing consumers with innovative and effective strategies for maintaining healthy and youthful skin [46].

This review embarks on a comprehensive exploration of the synergistic convergence between essential oils and nanoparticles in advanced dermocosmetic delivery systems. From

the challenges faced by essential oils to the transformative potential of plant nanovesicles, and the dynamic role of nanoparticles in skincare innovations, this synthesis illuminates the current landscape and future prospects of this intriguing intersection. Delving into realms of nanotechnology and sustainable cosmetic formulations, the promise of safe, effective, and environmentally conscious dermocosmetic solutions beckons, paving the way for a new era in skincare science.

## 2. Nanoparticles in Dermocosmetic Applications

In recent years, the field of dermocosmetic applications has witnessed significant advancements with the incorporation of nanoparticles, both inorganic and polymeric, revolutionizing cosmeceutical formulations (see Figure 1) [46]. Among the inorganic nanoparticles, titanium dioxide and zinc oxide have gained prominence for their multi-faceted roles in sunscreen formulations. These nanoparticles serve as physical blockers, forming a protective barrier on the skin surface that reflects and scatters harmful UV radiation, thus preventing sun damage and premature aging. Titanium dioxide and zinc oxide have demonstrated superior UV absorption capabilities, making them essential components in sunscreens and photoprotective formulations [47–50]. Moreover, their nanoscale dimensions offer advantages such as improved spreadability, reduced whitening effect, and enhanced adherence to the skin, addressing some of the limitations associated with conventional formulations [7,51].

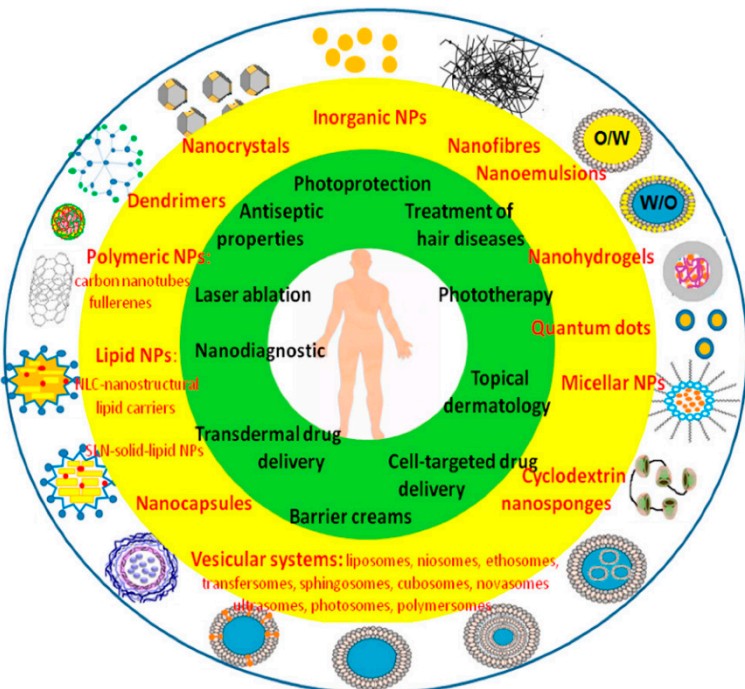

**Figure 1.** Illustration of various nanoparticles and their uses in dermatology for topical treatment. Reproduced with permission [46]. Copyright 2022, Multidisciplinary Digital Publishing Institute (https://creativecommons.org/licenses/by/4.0/), accessed on 25 February 2024.

Gold nanocarriers represent another intriguing aspect of nanoparticle application in dermocosmetics. These nanoparticles have garnered attention for their unique properties, including biocompatibility and ease of functionalization [52,53]. Gold nanocarriers serve as effective delivery systems, facilitating the controlled release of bioactive molecules [54]. The ability to encapsulate various active compounds, such as antioxidants and anti-aging agents, within gold nanocarriers enhances their stability and bioavailability [36,55]. This controlled release mechanism ensures a sustained and prolonged action of these active ingredients, contributing to the overall efficacy of cosmeceutical products [56,57].

Lipidic nanoparticles, including liposomes, solid lipid nanoparticles (SLNs), and nanostructured lipid carriers (NLCs), play a crucial role in cosmeceutical formulations by offering versatile solutions for delivering bioactive molecules to the skin [57,58]. Resembling cell membranes, liposomes act as reservoirs for bioactive molecules, enabling sustained release and minimizing systemic absorption [59]. Encapsulation of various compounds, such as benzophenone, glycolic acid, and curcumin, within liposomes has demonstrated improved cutaneous penetration and prolonged efficacy, particularly in anti-aging and antioxidant applications [57,60]. Additionally, SLNs and NLCs have emerged as effective carriers for both organic and inorganic sunscreens, providing occlusive properties and enhancing hydration while effectively blocking UV light [61–64]. Studies have indicated that these lipid nanoparticles exhibit superior UV absorption compared to conventional emulsions, thereby increasing the sun protection factor (SPF) of cosmeceutical products [65,66].

Polymeric nanoparticles also contribute to skin hydration and permeability, making them valuable in the prevention and treatment of wrinkles [67]. NLCs, in particular, have demonstrated their capability to promote skin hydration, attributed to their occlusive effect and the formation of a dense film upon application [68]. The nanometric size of particles ensures better coverage and uniformity on the skin, leading to improved hydration effects [47]. Additionally, the adhesive properties of NLCs make them suitable for incorporation into various pharmaceutical forms, including gels, creams, and lotions, further expanding their applications in dermocosmetics [57,65].

Table 1 offers a comprehensive summary of various nanomaterials utilized in dermocosmetic applications, showcasing their sizes, characteristics, and respective uses. Examples include gallic acid-coated gold nanoparticles (GA–AuNPs) renowned for antioxidant properties, zinc oxide nanoparticles in sunscreens for UV protection, and silver nanoparticles for antimicrobial activity in antidandruff shampoo. Moreover, niosomal carriers enhance drug permeation, copper oxide nanoparticles exhibit antimicrobial effects, and the rhein–phospholipid complex enhances solubility and skin permeability, all aiming to treat skin disorders effectively. Additionally, nanostructured lipid carriers loaded with curcumin improve skin permeation for conditions like psoriasis and acne, while solid lipid nanoparticles loaded with halobetasol propionate promise targeted drug delivery to minimize adverse effects.

**Table 1.** Examples of nanomaterials in dermocosmetic applications.

| Type of Nanomaterial | Size (nm) | Characteristics | Application | Ref. |
|---|---|---|---|---|
| Gallic Acid/Au NPs | 30.30 ± 3.98 | GA–AuNPs exhibit antioxidant properties, as they are evaluated as an anti-aging antioxidant | An ingredient with anti-aging properties aimed at rejuvenating and repairing the skin | [69] |
| Zinc oxide NPs | <30 | Zinc oxide nanoparticles primarily remain on the skin's surface, releasing zinc ions that penetrate superficial layers without significant cytotoxicity concerns, aligning with recent FDA safety guidelines | Application in sunscreens, providing effective UV protection while minimizing skin penetration and cytotoxicity risks | [70] |
| Silver NPS | ~40 nm and 13 | Silver nanoparticles demonstrated enhanced suspension stability against microbial contamination, suggesting their potential as an active ingredient in antidandruff shampoo formulations | Anti-Malassezia furfur activity | [71,72] |

**Table 1.** *Cont.*

| Type of Nanomaterial | Size (nm) | Characteristics | Application | Ref. |
|---|---|---|---|---|
| Niosomal carriers | 460 | Vesicle size depended on the surfactant mixture's hydrophile–lipophile balance, with drug incorporation influencing size and niosomes acting as effective enhancers for diclofenac sodium permeation across rabbit skin | Drug compartmentalization | [73] |
| Silica NPs | $291 \pm 9$ to $42 \pm 3$ | These particles demonstrated size-dependent uptake by skin cells, with positively charged particles showing enhanced cellular internalization, especially the smallest ones | Pharmaceutics and cosmetics applications | [74] |
| Copper oxide NPs | 61 to 69 | CuONPs exhibit potent antimicrobial properties against skin infection-causing microbes when combined with Thespesia populnea aqueous bark extract | Antimicrobial activity against skin-infection causing microbes | [75] |
| Rhein-phospholipid complex | $196.6 \pm 1.6$ | The rhein–phospholipid complex exhibit nano-sized particles and possess a high negatively charged surface. These nanoparticles show enhanced solubility, significantly improved skin permeability, and deep penetration into the skin | Topical formulation for treating skin disorders. | [76] |
| ZnO@CeO$_2$ nanostructures | 15 to 70 | One-dimensional rod-like ZnO@CeO$_2$ core@shell structures, synthesized with fine-tuned shell thicknesses with excellent optical absorption across both UV and visible regions | Optical stimuli-responsive in sunscreen cream | [77] |
| Lysine-Dendrimer | - | Unique three-dimensional structure that significantly reduces inflammation linked to acne without affecting non-acneic Cutibacterium acnes or commensal skin bacteria | Restore the microbiota balance in skin prone to acne | [78] |
| Curcumin loaded nanostructured lipid carriers | 96.2 | Curcumin-NLC are nanostructured lipid carriers (NLC) designed for topical delivery of curcumin, high entrapment efficiency ($70.5 \pm 1.65\%$), and significant improvement in skin permeation and retention compared to free curcumin formulations | Addressing persistent inflammatory conditions such as psoriasis and acne vulgaris caused by microbial activity | [79] |
| Halobetasol propionate-loaded solid lipid NPs | 200 | The solid lipid nanoparticles loaded with halobetasol propionate (HP-SLN) demonstrate promise as a delivery system for controlled drug release and targeted administration to the skin | Carrier for controlled drug release and targeted delivery to the skin, aiming to minimize adverse effects associated with clinical use, such as irritation, pruritus, and stinging | [80] |

Nanotechnological advancements stand out for their profound influence on the transformative landscape of dermocosmetic products. Delving into how these innovations have reshaped the skincare industry [57], dermocosmetic formulations incorporating nanotechnology have garnered considerable attention due to their ability to overcome traditional limitations, revolutionizing the way skincare and cosmetic products are developed and perceived. Nanoparticles, typically ranging from 1 to 100 nanometers in size, provide a platform for the controlled release of bioactive compounds, improved skin penetration, and enhanced stability of active ingredients [46,47,81]. Nanotechnological advancements have ushered in a new era for dermocosmetic products, offering solutions to traditional challenges and opening avenues for unprecedented formulations. From improving the stability and delivery of bioactive compounds to enhancing skin penetration and revolutionizing sunscreen formulations, nanotechnology has become a driving force in the evolution of skincare and cosmetics [82,83].

One of the key contributions of nanotechnology to dermocosmetics is the improved delivery of bioactive substances, such as antioxidants, vitamins, and peptides. Nanoparticles enable the encapsulation of these compounds, protecting them from degradation and promoting sustained release upon application. This controlled release mechanism not only enhances the stability of sensitive ingredients but also prolongs their interaction with the skin, maximizing therapeutic effects [57,84]. For example, encapsulating vitamins like C and E in nanocarriers protects them from oxidation, ensuring their potency and efficacy in combating oxidative stress and promoting skin health [85,86].

Liposomes, spherical vesicles composed of lipid bilayers, have been extensively explored in dermocosmetics for their ability to encapsulate a wide range of compounds, including both water-soluble and lipid-soluble ingredients [87–89]. Solid lipid nanoparticles, composed of lipids in a solid state, provide stability to incorporated actives and facilitate sustained release [90]. Polymeric nanoparticles, formed from biocompatible polymers, offer customization of release profiles and improved adherence to the skin [91]. Beyond enhancing stability and delivery, nanotechnology facilitates improved skin penetration, addressing the challenge of transporting active ingredients to deeper skin layers. Nanoparticles possess the capacity to overcome the skin's natural barrier, allowing efficient delivery of therapeutic compounds to targeted cells. This property is particularly beneficial for addressing skin conditions that require penetration beyond the superficial layers, such as in the case of anti-aging or dermatological treatments. Moreover, the nanoscale size of these carriers enhances their interaction with skin cells, ensuring optimal absorption and utilization of the encapsulated actives actives [57,92–96].

Traditional sunscreens often leave a white cast on the skin due to the larger particle size of these minerals [97,98]. Reducing the size of these particles not only removes the white residue but also enhances the even distribution of UV-blocking agents on the skin, thereby improving the overall effectiveness of sun protection [99]. However, it is crucial to address safety concerns associated with nanoparticle penetration through the skin, necessitating rigorous testing and regulation [51].

As research progresses, continued emphasis on safety and regulatory standards will be paramount to realizing the full potential of nanocosmetic innovations and ensuring consumer confidence in these transformative products.

## 3. Challenges of Essential Oils and Potential Solutions

In addition to their utilization in food, agriculture, and textiles, essential oils find applications across an array of industries, highlighting their versatility and efficacy (see Figure 2) [100]. Dermocosmetic applications particularly stand out, as essential oils are increasingly incorporated into skincare products for their multifaceted benefits, including soothing sensitive skin, combating acne, and promoting overall skin health. These oils serve as natural alternatives, offering consumers a holistic approach to skincare that aligns with growing preferences for organic and sustainable ingredients [5,101–103]. One major challenge lies in the volatility of essential oils, which refers to their tendency to evaporate easily

at room temperature. This characteristic can compromise the longevity of the fragrance and therapeutic effects of essential oils in cosmetic applications [104,105]. Essential oils are complex mixtures of volatile compounds, predominantly terpenoids, with varying boiling points. As a result, these volatile components can be lost through evaporation during the formulation process or upon application to the skin. This volatility not only affects the olfactory profile of the product but also hinders the maintenance of consistent concentrations of bioactive compounds required for therapeutic efficacy [101,106–109]. The low water solubility of essential oils constitutes another significant challenge in dermocosmetic formulations [110]. Most essential oils are hydrophobic, composed of lipophilic terpenes and aromatic compounds, making them poorly soluble in water. This poses obstacles in achieving homogenous dispersion within aqueous cosmetic formulations, as essential oils tend to separate from the water phase, leading to issues of poor stability and inconsistent delivery of bioactive components [111,112]. Moreover, the low water solubility limits the ease of incorporating essential oils into various cosmetic products, as their homogeneous distribution becomes a critical factor in ensuring uniform application and efficacy [113,114].

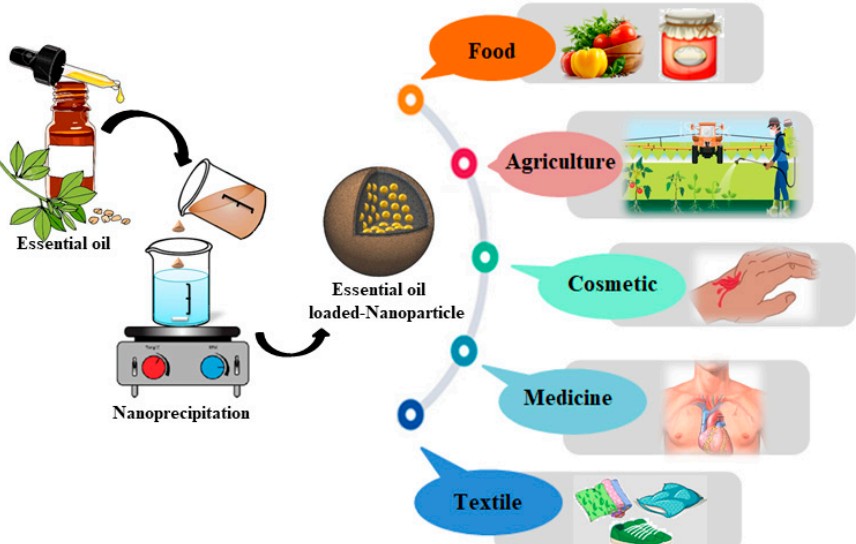

**Figure 2.** Various applications of essential oils encapsulated using the nanoprecipitation technique. Reproduced with permission [100]. Copyright 2020, Multidisciplinary Digital Publishing Institute (https://creativecommons.org/licenses/by/4.0/) accessed on 25 February 2024.

In addressing the challenges posed by essential oils' hydrophobicity and volatility, various strategies, including nanoencapsulation, become imperative. Among these, a promising approach involves harnessing plant nanovesicles for encapsulating and delivering essential oils, particularly in dermocosmetic applications [110,115–117]. Numerous techniques exist for nanoencapsulating essential oils into various systems, including emulsification, extrusion, nanoprecipitation, and complex coacervation (see Figure 3) [100,115,118]. This avenue signifies a significant advancement, aligning with the quest for enhanced stability and efficacy in diverse industrial applications. Plant nanovesicles, such as exosomes or extracellular vesicles, exhibit a hydrophobic character, mimicking the lipid composition of essential oils. This similarity facilitates their integration into cosmetic formulations, ensuring compatibility and reducing the risk of phase separation [116,119,120]. Moreover, the nanometric size of plant nanovesicles enhances their interaction with the skin, allowing for improved absorption and retention of essential oil components. This not only addresses the volatility concern by providing a reservoir for sustained release, but also contributes to a more efficient and controlled delivery of bioactive compounds [121].

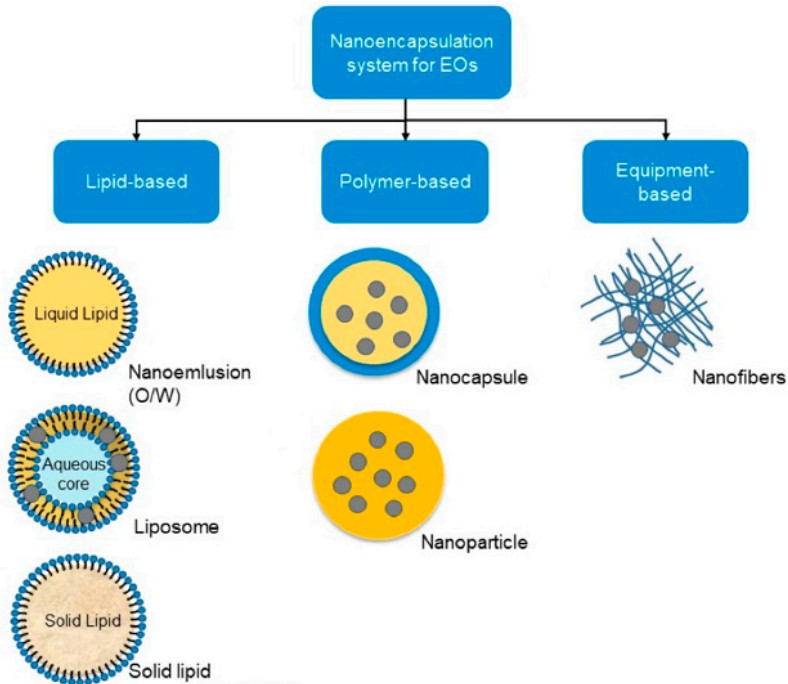

**Figure 3.** Different nanoencapsulation systems of essential oils. Reproduced with permission [118]. Copyright 2021, Multidisciplinary Digital Publishing Institute (https://creativecommons.org/licenses/by/4.0/), accessed on 25 February 2024.

In addition to the protective benefits provided by plant nanovesicles in dermocosmetic formulations, various essential oils offer valuable properties for skin health and wellness. Table 2 provides examples of essential oils commonly used in dermocosmetic applications, highlighting their distinct properties and versatile applications in cosmetic and pharmaceutical formulations.

The encapsulation of essential oils within plant nanovesicles serves as a protective barrier against the detrimental effects of environmental factors, such as light, oxygen, and temperature [116,117]. Essential oils are prone to degradation when exposed to these elements, leading to a loss of therapeutic efficacy [104]. Plant nanovesicles act as shields, preserving the integrity of the encapsulated essential oils and extending their stability. This is particularly crucial in dermocosmetic applications, where the maintenance of bioactivity over time is paramount for product effectiveness [122,123].

The lipid layers of these vesicles provide an ideal environment for incorporating lipophilic compounds, including the hydrophobic constituents of essential oils [124,125]. Passive and active loading techniques can be employed to enhance the encapsulation efficiency, ensuring that a higher percentage of essential oil components are successfully entrapped within the nanovesicles. This not only improves the homogeneity of essential oil dispersion but also facilitates their incorporation into various cosmetic formulations, expanding the range of dermocosmetic products that can benefit from the bioactive properties of essential oils [58,116].

The plant nanovesicles also present opportunities for targeted delivery in dermocosmetic applications [126]. They can be functionalized or modified to enhance their targeting abilities, allowing for specific delivery of essential oil components to distinct skin layers or cell types [58,116]. This targeted approach not only improves therapeutic outcomes but also reduces the risk of adverse effects, as the bioactive compounds are directed to their intended sites of action [127,128].

Despite the potential advantages of plant nanovesicles in dermocosmetic applications, several considerations need attention [110]. Standardized procedures for isolation, physico-chemical characterization, and stability evaluation are essential to ensure the reproducibility and scalability of the process [129]. Additionally, further research is needed to optimize

loading efficiency, particularly when dealing with the encapsulation of essential oils with varying chemical compositions [112,124,130]. Regulatory aspects, including sterility for intravenous administration, must be addressed to facilitate the regulatory approval of these innovative bionanosystems in the cosmetic industry [116,131].

**Table 2.** Examples of essential oils in dermocosmetic applications.

| Essential Oil | Properties | Application | Ref. |
|---|---|---|---|
| Lavender, tea tree, and lemon | Antimicrobial activity | Cosmetic preservative systems | [132] |
| Menthol | Menthol exhibits various biological activities, including antibacterial, antifungal, antipruritic, anticancer, and analgesic effects, as well as acting as an effective fumigant | Medicinal products for its cooling and biological effects | [133] |
| *Thymus vulgaris* L. (Thyme) | Hepatoprotective properties and to have effectiveness as expectorant agent, anti-acne agent, and as fungicidal and antiviral drug | Dermocosmetic and pharmaceuticals products | [134–136] |
| Citronella | Various activities such as antimicrobial, anthelmintic, antioxidant, anticonvulsant, antitrypanosomal, and wound healing properties, in addition to its mosquito repellent action | Pharmaceuticals, biomedical applications, cosmetics, food, veterinary, and agriculture applications | [137,138] |
| Rosemary (*Rosmarinus officinalis* L.) | Antioxidant, anti-inflammatory, antimicrobial, memory enhancement, digestive aid, hair and scalp health, pain relief, etc. | Gels, shampoos, soaps, rosemary water, cleansing milk, deodorant, anti-wrinkle cream, aftershave lotion, hydrating facial cream, cream for the eye contour area, etc. | [139–142] |
| Lavender (*Lavandula angustifolia* L.) | Antimicrobial, anti-inflammatory, healing, relaxing and calming, antioxidant, and analgesic properties | Dermocosmetic and pharmaceuticals products | [143,144] |
| *Tagetes minuta*, *Euphorbia granulata* and *Galinsoga parviflora* | Anti-inflammatory, antimicrobial, antiviral, and antioxidant properties | Dermocosmetic and pharmaceuticals products | [145] |
| Argan oil nanocapsules containing naproxen | Moisturizing, anti-aging, nourishing, anti-inflammatory, wound healing, hair care, UV protection, and antimicrobial properties | Cosmetic and transdermal local applications | [146–148] |

## 4. Addressing Skin Aging with Nanomaterials and Essential Oils

The integration of nanomaterials into the therapeutic efficacy of essential oils introduces innovative solutions for addressing skin aging and promoting tissue rejuvenation within dermocosmetics. Their inherent properties work synergistically, offering cutting-edge approaches to alleviate visible signs of aging [57,149]. Skin aging, marked by a decline in vitality, structural integrity, and moisture retention, encounters formidable resistance through this collaboration, marking a transformative phase in skincare innovation [150–152].

One notable application of this synergy involves integrating nanomaterials into formulations infused with essential oils, capitalizing on their combined potential to target specific skin layers effectively [153,154]. Nanocarriers, ranging from liposomes to solid lipid nanoparticles, act as efficient vessels for encapsulating bioactive compounds present in essential oils, ensuring their potency while facilitating precise delivery to the skin [155,156]. When encapsulated within these nanocarriers alongside essential oils, peptides renowned for stimulating collagen synthesis penetrate deeper skin layers, initiating a rejuvenating cascade that enhances skin elasticity and resilience [57,67].

Wrinkles, which symbolize the aging process of the skin, succumb to the therapeutic abilities of nanomaterials and essential oils, which work together to deliver agents that boost collagen and antioxidants [18,57,157]. Peptides like palmitoyl pentapeptide-4, encapsulated within nanocarriers combined with essential oils, penetrate deep into the skin, stimulating collagen synthesis and reducing oxidative stress, which is a key factor in premature aging.

Through meticulous release mechanisms, these compounds synergistically combat wrinkle formation, presenting a holistic anti-aging approach [158]. Moreover, dry skin, which is common in aging, encounters a significant challenge addressed by nanomaterials enriched with essential oils. These formulations encapsulate moisturizing agents such as hyaluronic acid, leveraging nanotechnology to improve their penetration and retention in the skin [37,56,57]. Expertly formulated through the application of nanotechnology, these emulsions create a delicate, moisturizing barrier atop the skin, alleviating dryness and delivering revitalization [81]. Moreover, the synergy of nanomaterials and essential oils introduces adaptive skincare formulations tailored to the evolving needs of aging skin [8,81]. pH-responsive nanocarriers, for example, regulate the gradual release of active ingredients according to changes in the skin's pH levels, thereby maximizing effectiveness. This adaptability proves crucial in navigating the multifaceted terrain of aging skin, where various factors converge to shape its ever-evolving landscape [159,160]. However, it is essential to acknowledge the challenges surrounding the utilization of nanomaterials in skincare. Rigorous safety assessments and transparent communication regarding their incorporation are imperative to foster consumer trust and responsible innovation [161,162].

In conclusion, the synergy of nanomaterials with essential oils represents a paradigm shift in dermocosmetics, offering multifaceted strategies to combat skin aging and promote tissue rejuvenation. From targeted delivery of collagen-boosting peptides to encapsulation of hydrating agents for dryness relief, this symbiotic alliance epitomizes skincare innovation. As understanding of nanotechnology advances, so too does its potential to redefine anti-aging skincare, promising healthier, more resilient, and youthful skin in the future.

## 5. Essential Oils and Nanoparticles for Advanced Dermocosmetic Delivery Systems

The transformation in dermocosmetic formulations occurs at the intersection of essential oils (EOs) and nanoparticles, providing insight into their intricate collaboration [18,47]. Nanoemulsions, characterized by droplets at the nanometer scale, emerge as essential carriers, showcasing enhanced stability, minimal toxicity, and outstanding compatibility with biological systems [59,163]. This synthesis delves into the scientific complexities, encapsulating significant discoveries and outlining both the present status and future prospects of essential oil–nanoparticle cooperation for advanced dermocosmetic delivery systems [57].

### 5.1. Precision Delivery Enabled by Nanoemulsions

Nanoemulsions serve as precision vehicles for synergizing essential oils with nanoparticles, offering a robust platform for encapsulating both hydrophilic and lipophilic active compounds [112,164]. This encapsulation tackles solubility challenges, enhances stability, and ensures optimal bioavailability. They act as guardians, mitigating the volatility of essential oils, thereby ensuring sustained efficacy [114,165,166].

The encapsulation process within nanoparticles involves a meticulous interplay of physicochemical properties, enabling controlled release mechanisms. These mechanisms extend the shelf life of essential oils and facilitate their controlled and targeted delivery [167,168]. Nanoemulsions emerge as avant-garde carriers in advanced dermocosmetic formulations due to their ability to deliver active compounds precisely.

The diminutive size of nanoparticles enables targeted delivery precision, navigating the intricate layers of the skin with unparalleled accuracy [169,170]. Essential oil compounds encapsulated within nanoparticles exhibit specific tropism towards distinct skin layers, optimizing therapeutic outcomes. Rigorous scientific investigations substantiate this targeted approach, providing evidence of enhanced permeation and efficacy [171–173].

Scientific literature abounds with examples illustrating the prowess of essential oil–nanoparticle formulations in specific dermocosmetic applications. Nanostructured carriers laden with essential oils like peppermint and rosemary showcase heightened efficacy in stimulating hair growth, attributing this phenomenon to the precision enabled by nanoparticles in traversing the skin layers [5,36,101,110,174].

### 5.2. Sustained Release Dynamics

Sustained release dynamics embedded in nanoparticles constitute a cornerstone in essential oil–nanoparticle synergy. These dynamics yield prolonged and controlled release of active ingredients, ensuring enduring skincare benefits [18,175,176]. Rigorous scientific studies elucidate how sustained release mechanisms optimize therapeutic effects while minimizing potential side effects, aligning seamlessly with the stringent safety demands of skincare products [177–179]. Scientific exploration into sustained release benefits extends to diverse essential oil–nanoparticle formulations [156]. Controlled release becomes imperative in addressing specific dermatological concerns comprehensively, presenting a transformative avenue for mitigating various skin-related challenges with precision and efficacy [153].

### 5.3. Current Scientific Landscape and Futuristic Trajectories

The current scientific landscape in dermocosmetics reflects a combination of natural active compounds from plants, into nanoemulsions. Scientific insights underscore the moisturizing and photoprotective properties of these formulations [180]. The trajectory, guided by scientific rigor, seeks to refine the synergy between essential oils and nanoparticles, unveiling optimized combinations and novel nanostructures tailored to specific skin needs [181,182].

Future trajectories in dermocosmetics pivot on scientific exploration, emphasizing new formulations and advanced delivery systems. Ongoing scientific research endeavors to unravel the intricacies of essential oil–nanoparticle interactions, aiming for heightened efficacy, safety, and multifunctionality. Scientific literature anticipates a new era in skincare, where meticulously crafted formulations offer targeted, sustained, and scientifically enriched benefits. The synthesis of essential oil–nanoparticle synergy represents a scientific milestone in dermocosmetic delivery systems. The integration of essential oils' nuanced properties with the precision of nanoparticles has given rise to scientifically validated formulations meeting modern consumer demands. As the industry pivots towards this transformative synergy, scientific exploration propels the current trajectory and future potential, promising a new era in skincare rooted in advanced formulations backed by robust scientific evidence.

To illustrate the advantages and limitations of plant nanovesicles compared to more classical nanoparticle systems such as liposomes and nanoemulsions, Table 3 is provided in this review. This table summarizes key characteristics including biocompatibility, targeting and delivery capabilities, stability, scalability, sustainability, complexity of production, cost, drug loading capacity, and stability. This comparison highlights the unique attributes of plant nanovesicles and provides insights into their potential applications in dermocosmetic delivery systems.

**Table 3.** Comparative analysis of plant nanovesicles, liposomes, and nanoemulsions in dermocosmetic delivery.

| Characteristic | Plant Nanovesicles | Liposomes | Nanoemulsions | Ref. |
|---|---|---|---|---|
| Biocompatibility | High | High | Variable | [183] |
| Targeting and delivery | Yes | Yes | Yes | [184] |
| Stability | Moderate | Variable | Variable | [185] |
| Scalability | Yes | Yes | Yes | [186] |
| Sustainability | Yes | Depending on their source | Depending on their source | [187–189] |
| Complexity of production | High | Moderate | Moderate | [190] |
| Cost | Moderate | High | Moderate | [191] |
| Drug loading capacity | Moderate | High | High | [192–194] |
| Storage stability | Variable | Moderate | Moderate | [195] |

## 6. Sustainability Considerations in Nanotechnology-Based Dermocosmetic Formulations

In navigating the landscape of nanotechnology-enabled dermocosmetic formulations, a profound emphasis on sustainability becomes imperative, transcending mere technological advancements [8,162]. While the realms of nanotechnology offer unprecedented avenues for enhancing skincare efficacy, the pursuit of sustainability within this domain remains integral for fostering safe, effective, and environmentally conscious dermocosmetic solutions [196,197].

Within the tapestry of nanotechnology, the utilization of sustainable practices emerges as a cornerstone of progressive skincare science. This paradigm shift encompasses a holistic approach, encompassing the entire lifecycle of dermocosmetic products—from formulation to disposal [198,199]. Key considerations encompass the judicious selection of raw materials, eco-friendly manufacturing processes, and the reduction of environmental footprints throughout production [200,201].

Central to this discourse is the integration of biodegradable materials into nanotechnological frameworks, encapsulating the essence of sustainability within cosmetic formulations [6,8]. By leveraging natural compounds and renewable resources, such as plant-derived nanovesicles, the cosmetic industry embarks on a transformative journey towards ecological harmony [191]. These biocompatible carriers not only enhance the efficacy and stability of dermocosmetic products but also epitomize a commitment to environmental stewardship [116,191,202].

Furthermore, the ethos of sustainability extends beyond the laboratory confines to encompass broader societal and ecological dimensions. It encompasses ethical considerations, such as fair trade practices and biodiversity preservation, ensuring that skincare formulations resonate with principles of social responsibility and ecological integrity [203–205]. Moreover, the promotion of circular economy principles encourages the repurposing and recycling of packaging materials, minimizing waste and fostering a regenerative skincare ecosystem [206,207].

As the cosmetic industry traverses the precipice of innovation, the integration of sustainability and nanotechnology heralds a new era in skincare science. It beckons a future where skincare products not only nurture the skin but also nurture the planet, embodying a harmonious synergy between human well-being and environmental preservation [8,208].

The journey towards sustainable cosmetic formulations underscores a transformative shift in skincare paradigms. By infusing nanotechnology with principles of sustainability, the cosmetic industry forges a path towards safe, effective, and environmentally conscious dermocosmetic solutions, catalyzing a renaissance in skincare science.

## 7. Conclusions

In the dynamic realm of dermocosmetic formulations, the convergence of essential oils (EOs) and nanoparticles represents a groundbreaking synergy that has the potential to redefine skincare. This scientific review delves into the intricate interplay between EOs and nanoparticles, particularly nanoemulsions, shedding light on their collaborative prowess in addressing various skin concerns and advancing the field of dermocosmetic delivery systems.

The essence of this transformative synergy lies in the precision and versatility offered by nanoemulsions as carriers for EOs. These nano-scale vehicles serve as guardians of bioactive compounds, encapsulating both hydrophilic and lipophilic actives with finesse. The encapsulation process not only enhances the stability of EOs but also addresses solubility challenges, ensuring optimal bioavailability and mitigating the volatility that often hinders sustained efficacy.

Wrinkles, a prominent sign of aging, find a formidable adversary in this synergy. Nanomaterials, such as palmitoyl pentapeptide-4 encapsulated in nanoemulsions, penetrate the skin effectively, stimulating collagen synthesis and promoting skin elasticity. Antioxidant-loaded nanoemulsions counteract oxidative stress, providing a controlled and

sustained release of cosmeceuticals to combat premature aging comprehensively. Moreover, the nanoemulsions contribute to the development of lightweight moisturizers that, with their fine droplets, create a smooth and hydrating layer on the skin, addressing the issue of dryness and contributing to a more youthful complexion.

The regenerative potential of nanomaterials is exemplified through stem cell-derived nanovesicles, offering a novel approach to skin rejuvenation. Laden with bioactive molecules and growth factors, these nanovesicles modulate cellular processes, stimulate collagen synthesis, and promote tissue repair. The targeted delivery precision of nanoparticles optimizes therapeutic outcomes, navigating through intricate skin layers with unparalleled accuracy. This adaptability is particularly crucial in addressing the varying conditions of aging skin, where factors like hormonal changes, environmental stressors, and metabolic shifts contribute to the complex aging process.

The sustained release dynamics embedded in nanoparticles constitute a cornerstone in essential oil–nanoparticle synergy, yielding prolonged and controlled release of active ingredients. This not only optimizes therapeutic effects but also aligns seamlessly with the stringent safety demands of skincare products. Rigorous scientific studies support the effectiveness of essential oil–nanoparticle formulations in specific dermocosmetic applications, demonstrating their transformative potential in mitigating various skin-related challenges with precision and efficacy.

The current scientific landscape in dermocosmetics reflects an integration of natural actives into nanoemulsions, emphasizing the moisturizing and photoprotective properties of these formulations. Future trajectories pivot on scientific exploration, seeking to refine the synergy between essential oils and nanoparticles. Ongoing research endeavors to aim for optimized combinations and novel nanostructures tailored to specific skin needs, anticipating a new era in skincare where meticulously crafted formulations offer targeted, sustained, and scientifically enriched benefits.

This transformative synergy, backed by robust scientific evidence, meets modern consumer demands for advanced formulations. As the industry embraces this paradigm shift, propelled by scientific exploration, it ushers in a new era in skincare characterized by precision, efficacy, and multifunctionality. The future holds promise for skincare formulations that cater to specific skin needs, rooted in the harmonious collaboration of essential oils and nanoparticles.

**Author Contributions:** Z.A-T., R.E.A., A.A., K.B., A.T. and R.A. contributed to conceptualization and writing—review and editing; Z.A.-T. contributed to formal analysis and writing—original draft; R.T. contributed to investigation and correction; Z.A.-T., R.A. and R.T. contributed to validation and finalizing the manuscript. All authors have read and agreed to the published version of the manuscript.

**Funding:** This research received no external funding.

**Data Availability Statement:** Not applicable.

**Conflicts of Interest:** The authors declare no conflicts of interest.

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
