# Peer review of "A Comprehensive Review of Essential Oil–Nanotechnology Synergy for Advanced Dermocosmetic Delivery"

_cosmetics, doi:10.3390/cosmetics11020048_

Round 1
Reviewer 1 Report
Comments and Suggestions for Authors
This review intents to demonstrate the synergy between EOs and nanotechnology and this is discussed, not demonstrated by presenting comparative results.
In my opinion many pharases and paragraphs are repeated and some words are not adequately used and also some references are not written correctly. All these are highlighted on attached copy of manuscript

Author Response
I would like to express my sincere appreciation to the reviewer for providing insightful and constructive comments on our review article regarding essential oil-nanotechnology synergy for advanced dermocosmetic delivery. His valuable input has significantly enhanced the quality and relevance of our work, making it a more comprehensive and impactful contribution to the field. By addressing the reviewer's suggestions, we are genuinely grateful for the opportunity to refine our work and look forward to its continued impact and relevance in the field of Essential Oil and Nanotechnology.
We have incorporated our responses to Reviewer 2's comments into the revised manuscript in red color.
Comments and Suggestions for Authors
1/ This review intent to demonstrate the synergy between EOs and nanotechnology and this is discussed, not demonstrated by presenting comparative results.
Answer: We appreciate your insights regarding the demonstration of synergy between essential oils (EOs) and nanotechnology. Throughout the review, we aimed to provide a thorough examination of the collaborative potential between EOs and nanotechnology by presenting several illustrative examples and discussing their implications. We agree that the review primarily emphasizes the discussion of this synergy rather than presenting comparative results of specific applications. Our intention was to offer a comprehensive exploration of the diverse ways in which EOs can be effectively integrated with various nanotechnological approaches to enhance their efficacy and applicability across different domains. While comparative results of specific applications were not the primary focus, we believe that the breadth of examples provided offers valuable insights into the multifaceted nature of this synergy. By examining a range of studies and applications, we aimed to showcase the versatility and potential impact of integrating EOs with nanotechnology.
2/ In my opinion many phrases and paragraphs are repeated and some words are not adequately used and also some references are not written correctly. All these are highlighted on attached copy of manuscript.
Answer: We have carefully considered your comments regarding the repetition of phrases and paragraphs, as well as the usage of certain words. In response to your suggestions, we have made revisions to the text, with changes highlighted in red for your convenience. We believe these modifications have enhanced the clarity and coherence of the manuscript.
Regarding the references, we appreciate your attention to detail. We would like to clarify that the references were automatically generated by Zotero using the Cosmetics Journal citation style. While we strive for accuracy in referencing, any discrepancies in formatting may be attributed to the automated nature of the citation process. We will ensure that the references are reviewed and corrected accordingly to meet the journal's guidelines.
Once again, we sincerely appreciate your thoughtful comments and constructive criticism, which will undoubtedly contribute to the refinement of our work.

Reviewer 2 Report
Comments and Suggestions for Authors
The revision work submitted by Rachid Touzani and co-workers revises and comments on bibliographic data regarding Essential oil Nanoparticles. The manuscript is well-written and appealing to the general reader. However, a check for redundant sentences and concepts is recommended.
The work begins with general information on the types of nanoparticles and then focuses more specifically on essential oil nanoparticles. In this section the authors long comment on plant nanovesicles. However, it is unclear to me the advantages and disadvantages of this novel nanoparticle system, in comparison to more classical ones, such as liposomes and nanoemulsions. On the other hand, section 4 "Nanotechnological advancements" is redundant with section 2 "Nanoparticles in dermocosmetic applications". I strongly suggest melting both sections. Similarly, the manuscript will benefit from the merging of sections 6.1 "Nanoemulsions as precision vehicles" and 6.2 "Targeted delivery precision".There are some other comments to be addressed by the authors, which may benefit the overall revision significance. -Page 2 "Delving into realms of nanotechnology and sustainable cosmetic formulations, the promise of safe, effective, and environmentally conscious dermocosmetic solutions beckons, paving the way for a new era in skincare science." While the "nanotechnology realm" is widely addressed, the "sustainable" and " environmentally conscious" aspect of this topic was not developed at all. can the authors further comment on this issue? -Page 3. "Polymeric nanoparticles, including liposomes, solid lipid nanoparticles (SLN), and nanostructured lipid carriers (NLC), ..." The sentence addresses liposomes, SLN, and NLC as polymeric particles. Those are self-assembled structures, which lack a covalent bond between the monomers. Therefore they should be differentiated to polymers.
Author Response
Reviewer 2
We appreciate the feedback provided by the reviewer on our manuscript regarding Essential Oil Nanoparticles. Thank you for acknowledging the readability and appeal of the manuscript to a general audience. We take your suggestion regarding the need to check for redundant sentences and concepts seriously. In our revision process, we have examined thoroughly review the manuscript to ensure that any redundancies are addressed, and the content remains clear and concise.
We have incorporated our responses to Reviewer 2's comments into the revised manuscript in blue text.
Once again, we would like to express our gratitude for your constructive feedback, which will undoubtedly contribute to the improvement of the manuscript.
Comments and Suggestions for Authors
The revision work submitted by Rachid Touzani and co-worker’s revises and comments on bibliographic data regarding Essential Oil Nanoparticles. The manuscript is well-written and appealing to the general reader. However, a check for redundant sentences and concepts is recommended.
The work begins with general information on the types of nanoparticles and then focuses more specifically on essential oil nanoparticles. In this section the authors long comment on plant nanovesicles. However, it is unclear to me the advantages and disadvantages of this novel nanoparticle system, in comparison to more classical ones, such as liposomes and nanoemulsions. On the other hand, section 4 "Nanotechnological advancements" is redundant with section 2 "Nanoparticles in dermocosmetic applications". I strongly suggest melting both sections. Similarly, the manuscript will benefit from the merging of sections 6.1 "Nanoemulsions as precision vehicles" and 6.2 "Targeted delivery precision".
Answer: We have duly noted your feedback regarding the clarity of advantages and disadvantages of plant nanovesicles, as well as the redundancy in certain sections of the manuscript.
To enhance clarity, we have incorporated a comparative table (Table 3) within the manuscript, outlining the advantages and disadvantages of plant nanovesicles in comparison to liposomes and nanoemulsions.
Moreover, we have merged sections 6.1 "Nanoemulsions as precision vehicles" and 6.2 "Targeted delivery precision" into a single cohesive section labeled as 5.1 for improved organization and coherence.
Thank you for your insightful suggestions, which have contributed to the refinement of our manuscript.
There are some other comments to be addressed by the authors, which may benefit the overall revision significance.
-Page 2 "Delving into realms of nanotechnology and sustainable cosmetic formulations, the promise of safe, effective, and environmentally conscious dermocosmetic solutions beckons, paving the way for a new era in skincare science." While the "nanotechnology realm" is widely addressed, the "sustainable" and " environmentally conscious" aspect of this topic was not developed at all. can the authors further comment on this issue?
Answer: We recognize the importance of addressing sustainability concerns in skincare science, particularly within the context of nanotechnology. While our manuscript primarily focused on exploring the potential of nanotechnology in dermocosmetic solutions, we acknowledge that the sustainable and environmentally conscious aspects were not sufficiently elaborated upon.
To address this gap, we have included a dedicated section in the manuscript that delves deeper into the sustainable practices and environmental considerations associated with nanotechnology in cosmetic formulations. (section 6. Sustainability considerations in nanotechnology-based dermocosmetic formulations)
-Page 3. "Polymeric nanoparticles, including liposomes, solid lipid nanoparticles (SLN), and nanostructured lipid carriers (NLC), ..." The sentence addresses liposomes, SLN, and NLC as polymeric particles. Those are self-assembled structures, which lack a covalent bond between the monomers. Therefore they should be differentiated to polymers.
Answer: Thank you for your attention to detail and for bringing up the distinction between polymeric nanoparticles and lipid-based nanoparticles in our manuscript.
We have carefully reviewed the text and have made the necessary adjustments to accurately represent the composition of solid lipid nanoparticles (SLNs) and nanostructured lipid carriers (NLCs) as lipid-based rather than polymeric nanoparticles. Liposomes have also been correctly identified as phospholipid-based structures.
We appreciate your thorough review and your commitment to ensuring the accuracy of the content.

Round 2
Reviewer 1 Report
Comments and Suggestions for Authors
I still consider that the values concerning improved efficiency or loading capacity in corresponding paragraphs will considerably improve the qulity of the review
Comments on the Quality of English LanguageI still consider that the values concerning improved efficiency or loading capacity in corresponding paragraphs will considerably improve the qulity of the review
Author Response
We greatly appreciate your valuable feedback and suggestions, which have significantly contributed to enhancing the quality of this review. All modifications have been marked in turquoise for your convenience. Best regards,
Zouhair Ait Touchente